



# Biological Nitrogen Fixation in CMIP6 Models

Taraka Davies-Barnard[1,2], Sönke Zaehle[2], Pierre Friedlingstein[1,3],

[1]College of Engineering, Maths, and Physical Sciences, University of Exeter, Exeter, UK
[2]Max Planck Institute for Biogeochemistry, Jena, Germany
[3]Laboratoire de Meteorologie Dynamique, Institut Pierre-Simon Laplace, CNRS-ENS-UPMC-X, Departement de Geosciences, Ecole Normale Superieure, 24 rue Lhomond, 75005 Paris, France

*Correspondence to*: T. Davies-Barnard (t.davies-barnard@exeter.ac.uk)

**Abstract.** Biological nitrogen fixation is the main source of new nitrogen into natural terrestrial ecosystems and consequently in the nitrogen cycle in many earth system models. Representation of biological nitrogen fixation varies, and
because of the tight coupling between the carbon and nitrogen cycles, previous studies have shown this affects net primary productivity. Here we present the first assessment of the performance of biological nitrogen fixation in models contributing to CMIP6 compared to observed and observation-constrained estimates of biological nitrogen fixation. We find that 9/10 models represent global total biological nitrogen fixation within the uncertainty of recent global estimates. However, 6/10 models overestimate the amount of fixation in the tropics, and therefore the extent of the latitudinal gradient in the global
distribution. For the SSP3-7.0 scenario of future climate change, models project increases in fixation over the 21st century of up to 80%. However, while the historical range of biological nitrogen fixation amongst models is large (up to 140 kg ha$^{-1}$ yr$^{-1}$ at the grid cell level and 43 - 208 TgN yr$^{-1}$ globally) this does not have explanatory power for variations in net primary productivity or the coupled nitrogen-carbon cycle. Models with shared structures can have significant variations in both biological nitrogen fixation and other parts of the nitrogen cycle without differing in their net primary productivity. This
points to systematic challenges in carbon-nitrogen model structures.

## 1 Introduction

The majority of earth system models (ESMs) of the latest generation that contribute to CMIP6 (Taylor et al., 2012) include a nitrogen cycle to better represent the terrestrial carbon cycle (Arora et al., 2020; Davies-Barnard et al., 2020). Nitrogen is a key nutrient requirement for plants to plants to take up carbon, and in its bioavailable inorganic form, is highly liable to
losses via gaseous and water processes (Thomas et al., 2013; Vitousek and Howarth, 1991). Over the last few decades, terrestrial carbon uptake has sequestered around a quarter of anthropogenic carbon emissions (Friedlingstein et al., 2020). However, previous assessments of ESMs have suggested that future projections of terrestrial carbon storage are decreased by 37 – 58% if nitrogen availability is accounted for (Wieder et al., 2015; Zaehle et al., 2014). Therefore, the accuracy of ESMs, which help guide policy on preventing further climate change, is partly determined by the functioning of the nitrogen
cycles within them.





The uptake of new carbon by plants is reliant on new sources of nitrogen, as existing nitrogen may not be bioavailable. The sources of this new input of nitrogen vary by biome, including anthropogenic inputs via addition of fertiliser 70 - 108 Tg yr$^{-1}$ (Lu and Tian, 2017; Potter et al., 2010) and increased deposition, and natural sources such as lightning $3.5 – 7$ TgN yr$^{-1}$ (Tie et al., 2002), atmospheric N deposition 63 TgN yr$^{-1}$ (Lamarque et al., 2013), weathering (Holloway and Dahlgren, 2002), and

biological nitrogen fixation (BNF) $40 – 141$ TgN yr$^{-1}$ (Davies-Barnard and Friedlingstein, 2020; Vitousek et al., 2013). In many natural ecosystems BNF is likely the largest natural or anthropogenic source of new nitrogen to the terrestrial biosphere. But because of the intricate processes that control fixation, and the lack of global estimates from observations, also the most uncertain (Meyerholt et al., 2016; Reed et al., 2011). Therefore, continued carbon sequestration in critical natural ecosystems that are present day and future carbon sinks is reliant on BNF. We need to know how well models are

representing current the quantity and distribution of BNF to assess the reliability of the functions and therefore the robustness of future projections of terrestrial carbon uptake. Studies of individual models suggest differences in representation of BNF can lead to widely differing future terrestrial carbon sequestration (Meyerholt et al., 2016; Peng et al., 2020; Wieder et al., 2015). Therefore inaccuracies in BNF representation could theoretically lead to errors in allowable emissions (Jones et al., 2013) for targets such as constraining warming to 1.5 or 2 °C (Millar et al., 2017).

BNF is performed by a large range of bacteria in virtually all parts of the terrestrial environment, including soil, litter, leaf canopy, decaying wood, and in association with bryophytes, lichens, and angiosperms (Davies-Barnard and Friedlingstein, 2020; Reed et al., 2011; Son, 2001; Tedersoo et al., 2018). BNF is frequently classified into symbiotic (higher plant associative) and free-living pathways (Cleveland et al., 1999; Reed et al., 2011). Symbiotic BNF makes up around two thirds of BNF and free-living BNF around one third (Davies-Barnard and Friedlingstein, 2020) or as much as 49 TgN yr$^{-1}$ (Elbert et

al., 2012).

Despite the complexity of BNF, most models have a simple BNF representation based on either: i) a linear relationship with net primary productivity (NPP) or ii) a linear relationship with evapotranspiration (ET), both derived from Cleveland et al., (1999) (see Table 1). However, recent analyses show that in non-agricultural biomes ET and NPP are poor predictors of both symbiotic and free-living BNF (Davies-Barnard and Friedlingstein, 2020; Dynarski and Houlton, 2018). Models with more

complex representations are mainly based on plant nitrogen demand, physiological limits, or optimality approaches (Fisher et al., 2010; Meyerholt et al., 2016; Wang et al., 2007) (see Table 1). While single model assessments have shown the importance of BNF to carbon sequestration, affecting the terrestrial carbon sink by up to a third (Meyerholt et al., 2016, 2020; Wieder et al., 2015), hitherto the performance of multiple models has not been assessed against observed BNF values.

## 2 Methods






## 2.1 ESM runs

We use results from 10 ESMs: CMCC-CM2, TaiESM1, CESM2, NorESM2, UKESM1, AWI-ESM, MPI-ESM, ACCESS, EC-Earth, and MIROC. The simulations used were the historical runs from CMIP6 deck simulations (Eyring et al., 2016) WCRP for the period 1950 – 2014. A list of the reference ids of the simulations used can be found in the SI.

## 2.2 BNF in the Models

A summary of the models' can be found in Table 1. Although there appears to be a range of approaches to BNF, every model considered here is partly or entirely based on Cleveland et al., (1999).

### 2.2.1 CABLE and CASACNP – Used in ACCESS

The Nitrogen cycle in the CABLE model (Law et al., 2017) of the ACCESS ESM relies on the CASACNP model, as
described by (Wang and Houlton, 2009; Wang et al., 2007). Symbiotic BNF is calculated as a function of soil moisture, soil temperature, soil N availability, and NPP. Free-living BNF is calculated using biome level observational averages adapted from Cleveland et al., (1999) with a range of 0.7 – 9.2 kg N ha$^{-1}$ yr$^{-1}$ (tropical forest highest, needleleaf forest lowest) (Wang and Houlton, 2009).

### 2.2.2 CLM4.5 – Used in CMCC-CM2 and TaiESM1

The Community Land Model version 4.5 (CLM4.5; Koven et al., 2013; Oleson et al., 2010) is used in the Euro-Mediterranean Centre on Climate Change coupled climate model (CMCC-CM2; Cherchi et al., 2019) and TaiESM1. The N component is described in Koven et al., (2013).

BNF is calculated as an exponential saturating function of NPP based on Thornton et al., (2007), which is based on Cleveland et al., (1999) with a 7 day lag to match seasonal BNF to NPP. There is no differentiation between symbiotic and
free-living BNF.

### 2.2.3 CLM5 – Used in CESM2 and NorESM2

The Community Land Model version 5 (CLM5; Lawrence et al., 2019) is used in The Community Earth System Model Version 2 (CESM2; Danabasoglu et al., 2020) and the Norwegian Earth System Model version 2 (NorESM2; Seland et al., 2020). CLM5 is the latest version of CLM and represents a suite of developments on top of CLM4.5. The N component is
described in Fisher et al., (2010); and Shi et al., (2016).

Symbiotic BNF is calculated on a carbon cost basis for acquiring N, derived from the Fixation and Uptake of Nitrogen (FUN) approach (Fisher et al., 2010). Free-living BNF in CLM5 is calculated separately as a function of evapotranspiration based on Cleveland et al., (1999) (Lawrence et al., 2019).





### 2.2.4 JSBACH – Used in MPI-ESM and AWI-ESM1

JSBACH version 3.20 model (Goll et al., 2017) is used in the Max Planck Earth System Model version 1.2 (MPI-ESM; Mauritsen et al., 2019) and Alfred Wegener Institute Earth System Model (AWI-ESM). The N component is described in Goll et al., (2017).

BNF is calculated as an exponential saturating function of NPP based on Peter E. Thornton et al., (2007), which is based on Cleveland et al., (1999). The BNF function is calibrated to produce 100 TgN yr$^{-1}$ with NPP of 65 Pg yr$^{-1}$ (Goll et al., 2017).
There is no differentiation between symbiotic and free-living BNF.

### 2.2.5 JULES – Used in UKESM1

The Joint UK Land Environment Simulator version 5.4 (JULES-ES; Best et al., 2011; Clark et al., 2011) is used in the UK Earth System Model (UKESM1; Sellar et al., 2020.). The N component is described in (Wiltshire et al., 2021) and Sellar et al., (2020).

BNF is calculated as a linear function of NPP, 0.00016 kg N per kg C NPP (Wiltshire et al., 2021), based on Cleveland et al., (1999). There is no differentiation between symbiotic and free-living BNF.

### 2.2.6 LPJ-GUESS – Used in EC-Earth

The Lund-Potsdam-Jena General Ecosystem Simulator version 4.0 (LPJ-GUESS; Olin et al., 2015; Smith et al., 2014) is used in the European community Earth-System Model (EC-Earth; Hazeleger et al., 2012). The N component is described in
Smith et al., (2014).

BNF is a linear function of ET, 0.0102 ET (mm yr$^{-1}$) + 0.524 (Smith et al., 2014), based on Cleveland et al., (1999). There is no differentiation between symbiotic and free-living BNF. The amount of BNF is capped at 20 kg ha$^{-1}$ yr$^{-1}$.

### 2.2.7 VISIT-e – Used in MIROC

VISIT-e is used in the Model for Interdisciplinary Research on Climate, Earth System version 2 for Long-term simulations
(MIROC-ES2L) (Hajima et al., 2020). The nitrogen component is described in Hajima et al., (2020).

BNF is a linear function of ET, based on Cleveland et al., (1999). Symbiotic and free-living BNF are calculated using the same function and distinguished by symbiotic BNF being directly available to plants, whereas free-living BNF is assumed to be part of the litter. Symbiotic BNF represents 50% of BNF. In cropland a higher level of BNF occurs for nitrogen fixing crops, but non-fixing crops have the same BNF as natural vegetation (Hajima et al., 2020).




## 2.3 Observations

Following the methods of Davies-Barnard & Friedlingstein, (2020) we reviewed the BNF literature to find observational data that covered all, or close to all, BNF at a field site (i.e. including symbiotic and free-living fixation of as many BNF types as are present). The locations of the site observations used can be found in SI Figure 1. Further details of the observations are in

SI Table 1. Few measurements are available, with studies usually focussing on either symbiotic or free-living BNF. Since recent meta-analysis suggests that free-living is approximately a third of total BNF, and higher in some regions, we only consider data that includes explicitly both symbiotic and free-living BNF or states that all sources of BNF are measured.

## 3 Present day BNF

The majority of the models have total global BNF between 80 to 130 TgN yr$^{-1}$ (Figure 1 a), within the uncertainties of two

recent observation-based BNF estimates (Davies-Barnard and Friedlingstein, 2020; Vitousek et al., 2013). There is little relationship between BNF function and total global BNF, with the two models using BNF based on ET encompassing the lowest and second highest values. There is, in some instances, as much variation in global total BNF within models that share components than between different models (see Methods). For instance, CESM2 and NorESM2 share the same land surface model and the modelled BNF is still 43 TgN yr$^{-1}$ different. The range between CMCC, TaiESM1, UKESM1, MPI-

ESM, and AWI-ESM which all calculate BNF from NPP is just 36 TgN yr$^{-1}$. This is suggestive of a substantial role for climate in modelling of BNF and a deliberate clustering to the most common BNF estimate (Davies-Barnard and Friedlingstein, 2020).

However, while for most ESMs the global BNF estimates show good agreement with Davies-Barnard & Friedlingstein, (2020) and Vitousek et al., (2013), too much BNF occurs in the tropics. In the low latitudes (30N to 30S) 6 of the 10 models

are above the observation-based estimate (Figure 1 b), in the mid latitudes only 1 model is above (Figure1 c) and in the high latitudes none (Figure 1 d). The multi-model mean of BNF from CMIP6 ESMs compared to an observation-based estimate (Figure 2) shows a broad agreement in spatial patterns, although there are clear weakness of the ESMs' BNF estimates is in tropical forests, where BNF is overestimated. This is to be expected, as most of the models are based on the data and linear modelling presented in Cleveland et al., (1999), which have  subsequently been revised  studies have revised to substantially

lower tropical forest BNF (Davies-Barnard & Friedlingstein, 2020; Sullivan et al., 2014; Vitousek et al., 2013). Although there are sources of error in the models, notably differing climate in the models' historical simulation compared to observed, these errors persist in the land surface model components of the ESMs when driven with observed data (see SI Figure 2).

The pattern of high BNF in the tropics is partly due to a small number of models with very high BNF (SI Figure 3). ACCESS has areas of anomalously high BNF in the tropics of up to 139 kg ha$^{-1}$ yr$^{-1}$. MIROC also has grid-cells of up to 193

kg ha$^{-1}$ yr$^{-1}$. Whereas in other models the tropical peak is below 41 kg ha$^{-1}$ yr$^{-1}$. While measurements of BNF from individual nitrogen fixating plants can be over 100 kg ha$^{-1}$ yr$^{-1}$, these rarely occur at a density of more than 30% cover (Davies-Barnard





and Friedlingstein, 2020). At the field scale BNF rarely exceeds around 15 kg ha$^{-1}$ yr$^{-1}$ for free-living BNF and 20 kg ha$^{-1}$ yr$^{-1}$ for symbiotic BNF (Davies-Barnard and Friedlingstein, 2020). Therefore, values much above 35 kg ha$^{-1}$ yr$^{-1}$ at an ESM grid-cell level seem improbable.

Comparison of models to individual BNF field-scale observations of all BNF (free-living and symbiotic) (Figure 3) show similar differences in latitudinal variation as the global and averaged data comparisons (Figures 1 and 2). The models underestimate mid latitude wetland and peatland BNF (Massachusetts and S Germany, Figure 3 b) (Schwintzer, 1983; Waughman and Bellamy, 1980) and desert BNF (Negev Desert Israel, Figure 3 b) (Russow et al., 2008). These locations show the systemic problem with BNF predicated on NPP and focused on symbiotic BNF. Although the NPP is relatively

low, the BNF is high due to the presence of free-living BNF (Russow et al., 2008; Schwintzer, 1983; Waughman and Bellamy, 1980). Free-living BNF is less likely to adhere to the assumption of being related to plant productivity, as by definition it is not directly associated with plants. Symbiotic BNF represents only 0.11 kgN ha$^{-1}$ yr$^{-1}$ in the Negev desert measurements, but the biological crusts fix 9 – 13 kgN ha$^{-1}$ yr$^{-1}$ (Russow et al., 2008). Considering only symbiotic BNF the models are on the correct order of magnitude. Unlike other observation sites, where some discrepancies between models and

observations can be partially attributed to differences in land cover, the models are capturing desert as a low productivity environment, with the NPP and ET based models all having very low BNF. The error therefore, is in the assumption that low productivity equates to low BNF. However, it is unclear at what time-scale free-living BNF becomes available to plants, and it remains uncertain to what extent it contributes to future carbon sequestration. Given that free-living BNF makes up 34 – 49% of total BNF, this suggests that in terms of BNF that is bioavailable to plants contributing significantly to NPP, the

modelled values ought to be lower than the global estimates shown above.

The low latitudes have a similar observed distribution of BNF to the mid latitudes, but the models generally have higher BNF, with 3 stark examples (Figure 3). These are all forest locations (Tierney et al., 2019; Zheng et al., 2016) with either tropical broadleaf or pine species and relatively high productivity environments. We can see from these locations, as well as the tropical forests of S Costa Rica (Sullivan et al., 2014) that the NPP based models are particularly liable to overestimations

of BNF in the tropics.

## 4 BNF in Future under SSP3-7.0

All the models simulate an increase in NPP over the 21$^{st}$ century in SSP3-7.0 due to the combined effects of rising atmospheric carbon dioxide and climate change. Given the constrained stoichiometric ratios within plants and soils, such an increase in productivity requires additional nitrogen to sustain growth. Work on the structural uncertainty to the carbon cycle

caused by BNF in individual models (Meyerholt et al., 2016, 2020; Wieder et al., 2015) indicates that changes in the representation of BNF and its assumed dependency on NPP, ET, or plant N demand lead to significant variation in carbon storage under elevated atmospheric carbon dioxide within the same model structure. In the context of these results and the





large range of present-day BNF simulated between CMIP6 models, it would be a logical corollary if the magnitude of simulated changes in NPP was associated with the magnitude of simulated change in BNF in the SSP3-7.0 scenario.

However, in this ensemble, the increases in BNF are not proportional to those in NPP (Fig, 4 a and b).

The models with BNF as a function of NPP should have BNF increases approximately commensurate with their increase in NPP (Figure 4 b). This is true for MPI and UKESM1, where relative changes in NPP and BNF fall nearly onto the 1:1 line. CMCC, which employs a similar representation, deviates from parity, because in parts of the tropics the simulated BNF is at the saturation-level of NPP and has reached the model prescribed maximum (see Table 1). TaiESM1, which uses the same

underlying land model as CMCC, shows a closer relationship between NPP and BNF. This is due to the lower tropical NPP in this model leading to the BNF being further from saturation point compared to CMCC. All these models suggest little change (relative to the whole model ensemble) in net N mineralisation or N loss (Figure 4 c), implying that the change in the terrestrial N budget is primarily driven by the NPP-related increase in BNF. The N deposition (where the model output is available) is very similar across these models as they should all have used the same prescribed boundary condition.

EC-Earth has the lowest BNF increase (1.3 TgN) over 2090-2100 compared to 2015-2025 under the SSP3-7.0 scenario, but a relatively high NPP increase (17.5 PgC) (Figure 4 d). Whereas the other ET driven model, MIROC, has the highest increase in NPP (22.0 PgC) and a BNF increase of 32.0 TgN (Figure 4 d). In the context of the whole ensemble, these two models have relatively high NPP given their change in BNF (in Figure 4 b they are below the 1:1 proportionality line). Both models also have the two largest increases in vegetation carbon to nitrogen ratios (Figure 4 d, EC-Earth + 12.9 and MIROC + 17.5,

in a model range of -29.3 to +17.5), probably because of a large fraction of vegetation carbon increase in woody biomass. This C:N ratio change effectively decreases the relative increase in demand for nitrogen associated with the increase in NPP, and illustrates that stoichiometry and BNF together affects the magnitude of the nitrogen constraint on terrestrial carbon storage (Meyerholt et al., 2020).

The two models using the FUN (Fisher et al., 2010) carbon cost function for BNF have almost identical absolute changes in

NPP, BNF, N dep, Nfert, and N loss (Figure 4 e and SI Figure4). They have the largest increase in BNF but proportionally less NPP change than BNF (they are above the 1:1 line in Figure 4 b). In effect, the extra supply of nitrogen via BNF in these models is not converting to an increase in NPP as efficiently as in the rest of the model ensemble. Despite their similar land model, CESM2 has the largest N uptake increase of all the models in the ensemble, whereas NorESM2 is the only model projecting a decrease in plant N uptake (346 TgN yr$^{-1}$ and -51 TgN yr$^{-1}$ respectively). This difference is likely related to

diverging projections in net N mineralisation, which is 788 TgN yr$^{-1}$ for CESM2 and 267 TgN for NorESM2 (ensemble range 75.8 TgN yr$^{-1}$ – 385.4 TgN yr$^{-1}$). In contrast, we see only 0.9 PgC yr$^{-1}$ difference in increase of NPP between these two models, in a model ensemble range of 1.3 – 22.6 PgC yr$^{-1}$ (Figure  4 a). From this we can see that in the underlying model, CLM5, nitrogen limitation plays only a small role in determining NPP and future terrestrial carbon sequestration. The large increase in N uptake in CESM2 compared to NorESM2 is not leading to proportional differences in NPP.

ACCESS-ESM shows a different pattern of changes in nitrogen model components to the rest of the model ensemble (Figure 4 e). Like CESM2 and NorESM2, ACCESS has proportionally less NPP increase for the amount of BNF increase, though



the absolute levels are much lower for both (Figure 4 b). It is possible that this is due to ACCESS including the phosphorus cycle in addition to the nitrogen cycle, and therefore, increases in NPP are not only constrained by the magnitude and increase in BNF, but also phosphorus availability. ACCESS is the only model where the C:N ratio change is not

approximately proportional compared to the rest of the ensemble to the change in NPP (Figure 4 e). Similarly, it has the largest increase in N loss.

## 5 Discussion

The historical simulations compared to data for BNF reveal a mixed message, with ensemble members generally performing well in high latitudes and at the global total scale, but poorly in the mid-latitudes and tropics. For the future scenario of

SSP3-7.0, the impact of the projected change in BNF over time under elevated atmospheric carbon dioxide and other environmental changes is more difficult to assess because of conflicting drivers and multiple biosphere interactions that confound the imprint of changing BNF on the terrestrial carbon cycle response.

Limitations in methodology of both observations and model simulations partly account for the lack of agreement between them. For models, the simulations are not specific to the site, but rather taken from the closest grid-cell of a global

simulation. Were site-level simulations with observed driving climate data available and the correct land cover (particularly vegetation) prescribed, it is possible models would perform better. For the data, the comparison is made with simple upscaled measurements grouped by biome, which is vulnerable to skew in the underlying data (Davies-Barnard and Friedlingstein, 2020). The underlying BNF data for the historical comparison also has substantial limitations. For instance, the most commonly used method of measuring BNF, acetylene reduction assay, requires calibration to avoid variation of up

to two orders of magnitude, that ~70% of studies fail to do (Soper et al., 2021). The literature is also biased away from null results, making an accurate understanding of the processes underlying BNF more difficult. Thus, the problems with model representation of BNF are symptomatic of wider uncertainties in BNF observations and upscaling.

The challenge for progressing BNF modelling is what would be a suitable replacement for the functions currently used. Symbiotic fixation is around two thirds of total BNF (Davies-Barnard and Friedlingstein, 2020) and is the focus of the more

process based models of BNF (as used in ACCESS, CESM2, and NorESM2). However, work with herbaceous legumes suggests that fixers may have little variation in whole plant biomass whether the nitrogen is fixed or provided as fertiliser, such that carbon cost of acquiring nitrogen symbiotically may be much lower than previously thought (Wolf et al., 2017). Attempts to establish an empirical relationship between BNF and climate or soil properties at macro scale have not indicated any robust relationship and biome based upscaled values have low data levels and high uncertainties (Davies-Barnard and

Friedlingstein, 2020). Abundance of fixers is an important parameter in the CESM2 and NorESM2 models and has a large impact on total fixation and response to fertilisation (Fisher et al., 2018), but in observations it is poorly constrained (Davies-Barnard and Friedlingstein, 2020) and not well correlated with total fixation rates, to the point of being anti-correlated



(Taylor et al., 2019). Free-living BNF makes up around a third of all BNF, is comprised of a heterogeneous set of organisms (Reed et al., 2011), making a single process based model challenging. Thus, the two CMIP6 models that account separately

for free-living BNF use static biome level upscaling based on data more than 20 years old, or a simple empirical relationship with evapotranspiration (see Table 1).

In the future scenarios, the multiple sources of uncertainty as to how and to what extent BNF will change make any definitive statements about the capacity of models to capture BNF changes difficult. While increased atmospheric carbon dioxide tends to increase BNF (Liang et al., 2016), nitrogen addition in the form of deposition or fertilisation tends to supress

BNF (Zheng et al., 2019), and effects from land use change (Zheng et al., 2020), increased temperature, reduced precipitation and other climate change  as well as the potential effects of climate induced land cover change that may alter the composition and location of biomes. It is challenging to predict which of these factors will predominate over the coming century.

Regardless of the change in BNF in future, it is revealing that while single parameter perturbation experiments suggest BNF significantly affects terrestrial carbon storage (Meyerholt et al., 2016; Wieder et al., 2015) when in a dynamic system the

effects of BNF are subsumed by structural differences in the nitrogen and carbon models, as well as the larger effects of increasing carbon dioxide.

## 5 Conclusions

BNF is an important part of the nitrogen cycle, and previous work has shown how nitrogen availability (Zaehle et al., 2014) and BNF in particular impacts terrestrial carbon storage (Meyerholt et al., 2016; Wieder et al., 2015). Here we have shown

that although there are shortcomings in the representation of BNF in CMIP6 models, BNF is not a direct control on future carbon uptake when considered in a dynamic system. Some models have a strong relationship between NPP and BNF, but the models that do not utilise changes in equally uncertain parts of the nitrogen cycle.

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

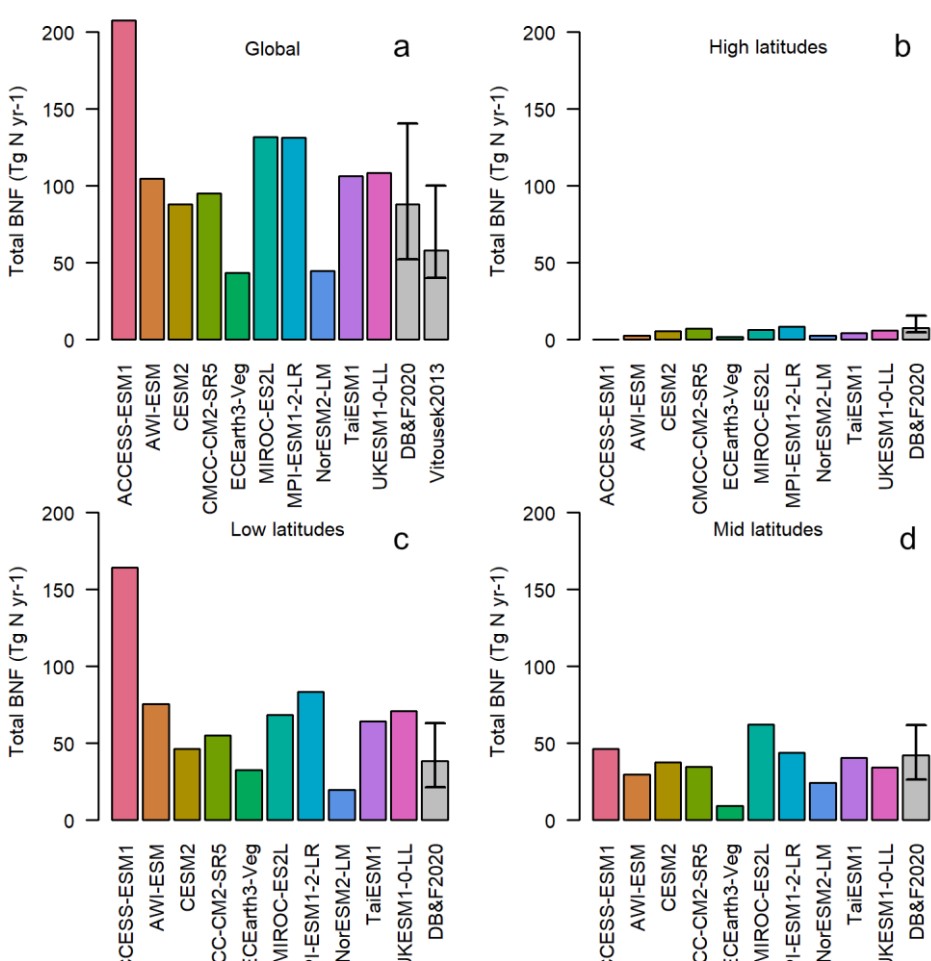

Figure 1: T The global total annual BNF (TgN yr⁻¹) for the average of the period 1980 – 2014. The grey bars areas represent the observationally-constrained ranges by Davies-Barnard and Friedlingstein, (2020) and Vitousek et al., (2013).



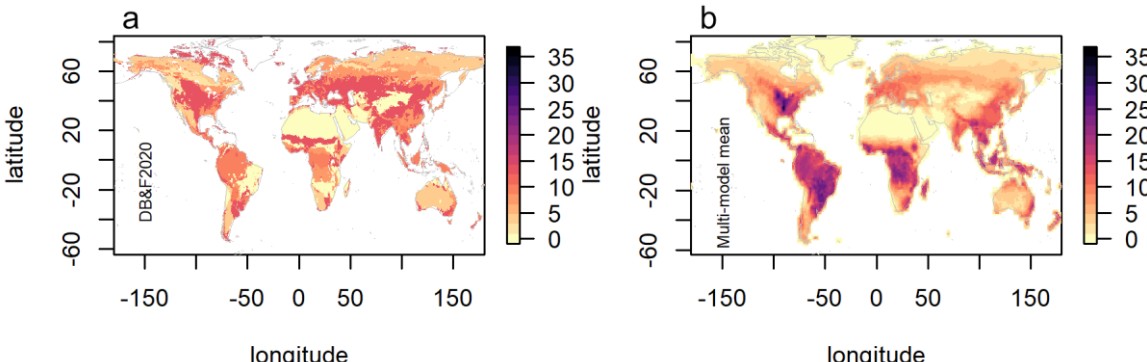

Figure 2: Map of observation-biome-based estimates of BNF (a) and a map of multi-model mean BNF for the period 1980 – 2015 (b).







Figure 3: BNF field-scale observations plotted over the model BNF value for the nearest grid-cell, matched to the latitude and longitude. Top panel: more than 60°N or 60°S. Middle panel: less than 60°N and more than 30°N less than 60°S and more than 30°S. Lower panel: observations less than 30°N and less than 30°S. The black lines represent single values, or the confidence range as reported by the paper.





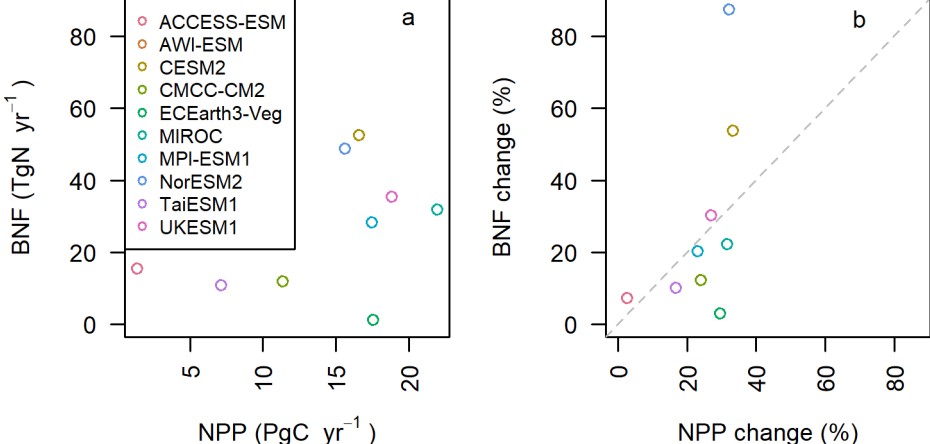

Figure 4: a and b show the change in BNF and NPP between the first and last decade of SSP370. c, d, and e show the
normalised changes in nitrogen components in the models. Each dot represents the normalised change in [the variable]
during the 21st Century, in SSP370. Each line represents a model, and each plot is a group of models that deal with BNF
similarly (from the top left: carbon cost or mechanistic, ET, and NPP).

**Table 1.** Summary of the model's BNF representations. The theoretical maximum BNF value refers to any limit imposed by
the equations in the model, e.g. a saturation point.

| ESM | LSM | Main driver | BNF representation | Theoretical maximum BNF value | Reference |
|---|---|---|---|---|---|
| CMCC-CM2 TaiESM1 | CLM4.5 | NPP | Non-linear function of NPP | 18 kgN ha$^{-1}$ yr$^{-1}$ | Oleson et al. (2013) |
| CESM2 NorESM2 | CLM5 | NPP (via C Cost function) & ET | Symbiotic N fixation according to the FUN model, Free-living N fixation linearly dependent on | None | Lawrence et al., (2019) |



| | | | evapotranspiration. | | |
|---|---|---|---|---|---|
| AWI-ESM MPI-ESM | JSBACH | NPP | Non-linear function of NPP | ~2235 kgN ha$^{-1}$ yr$^{-1}$ | Goll et al., (2017); Mauritsen et al., (2019) |
| UKESM1 | JULES-ES | NPP | Linear function of NPP, 0.0016 kg N per kg C NPP | None | (Wiltshire et al., 2021) |
| EC-Earth | LPJ-GUESS | ET | Linear function of ecosystem evapotranspiration, 0.102 cm yr$^{-1}$ ET +0.524 per kg N ha-1 | 20 kg N ha$^{-1}$ yr$^{-1}$ | Smith et al., (2014) |
| ACCESS | CABLE / CASACNP | NPP, soil temperature, soil moisture | Free-living BNF prescribed with no temporal variation from a combination of biome-based look-up. Symbiotic BNF process-based model. | Free-living BNF: 9.2 kgN ha$^{-1}$ yr$^{-1}$ Symbiotic: none | Law et al., (2017); Wang et al., (2007); Wang & Houlton, (2009) |
| MIROC | VISIT-e | ET | Linear function of ET. | None | Hajima et al., (2020) |