# Peer review of "Biological Nitrogen Fixation in CMIP6 Models"

_Biogeosciences, 2021_

## Referee Comment (RC1)

Overview

The work by Davies-Barnard and colleagues evaluates the Biological Nitrogen Fixation (BNF) simulated by CMIP6 models against observation. The authors also assess the influence of BNF on projected changes in the terrestrial carbon storage simulated by the CMIP6 models under the SSP370 scenario. This work presents the principal abilities and limitations of state-of-the-art earth system models in representing BNF fostering the future improvement of BNF in earth system models. At this stage, the manuscript requires some missing information and deeper discussion before being ready for publication, as described in the following sections.

General comments

The representation of BNF is divided between direct use of Cleveland et al. (1999) parameterization and models with more complex techniques. The first case, then, represents a well-established description of BNF, while the latter depicts the effort in improving the process-based representation of BNF. Are the models departing from the Cleveland et al. (1999) parameterization improving the BNF computation?
Similarly, does the separate modelling of symbiotic BNF and free-living BNF improve the simulated BNF?
A deeper discussion of these points would emphasize the effort needed to improve BNF representation in the current Earth System Models at global and local scales.

In the local comparison between model and observation, are there any differences in land cover type between model and observations, and among models? In fact, models with vegetation closer to the observed one could reach better results thanks to the land cover representation despite the BNF parameterization.

The authors should also present the land-only and future scenario simulations in the methodology section (section 2.1). Moreover, the authors should explain the choice of ssp370 as the future scenario. Why don't you analyze other future scenarios?

Specific comments

Line 21: should be Eyring et al., 2016 instead of Taylor et al., 2012.
Line 120: SI Table 1 is not available in the supplement link of the manuscript.
Figure 1: "T The global" remove the initial "T"
Figure 1: add information about the panels in the caption.
Figure 1: the second row has the name of the model truncated due to the figure caption. Reshape the figure.
Lines 139-140: "which […] tropical forest BNF". Rephrase.
SI Figure 2: second line in the caption "CM5" should be "CLM5"
SI Figure 2: Why do you compute the values over the 2005-2014 period, while the reference period is 1980-2014 in the paper?
SI Figure 3: explain the grey lines of panel a in the caption.

Figure 3, line 484: "reported by the paper". Which paper?

Figure 4: panels c, d and e are missing.

Lines 261-262: "Some models […] nitrogen cycle". Rephrase.

---

## Author Comment (AC1)

**Response to reviewer 1**

We thank the reviewer for their time in reviewing this paper. The reviewer comments are in *blue italics* and our responses are in plain black text.

*Overview*

*The work by Davies-Barnard and colleagues evaluates the Biological Nitrogen Fixation (BNF) simulated by CMIP6 models against observation. The authors also assess the influence of BNF on projected changes in the terrestrial carbon storage simulated by the CMIP6 models under the SSP370 scenario. This work presents the principal abilities and limitations of state-of-the-art earth system models in representing BNF fostering the future improvement of BNF in earth system models. At this stage, the manuscript requires some missing information and deeper discussion before being ready for publication, as described in the following sections.*

*General comments*

*The representation of BNF is divided between direct use of Cleveland et al. (1999) parameterization and models with more complex techniques. The first case, then, represents a well-established description of BNF, while the latter depicts the effort in improving the process based representation of BNF. Are the models departing from the Cleveland et al. (1999) parameterization improving the BNF computation? Similarly, does the separate modelling of symbiotic BNF and free-living BNF improve the simulated BNF? A deeper discussion of these points would emphasize the effort needed to improve BNF representation in the current Earth System Models at global and local scales.*

We deliberately avoid making a direct judgement on these issues, as while the physical process basis of the models using Cleveland et al. (1999) is weak, the more complex models are not necessarily performing better (e.g. ACCESS), particularly given the increased computational cost.

However, we see your point that there is a need to address this, and have added more direct comparisons of model performance and discussion of these issues to the Discussion and Conclusions.

*In the local comparison between model and observation, are there any differences in land cover type between model and observations, and among models? In fact, models with vegetation closer to the observed one could reach better results thanks to the land cover representation despite the BNF parameterization.*

While the models might be able to simulate BNF better if the vegetation closer to the observed, the reason we chose not to pursue this approach was that the difference in results of BNF between the LSMs driven by observations (including land cover) and their corresponding ESMs is small. Therefore, we would not expect the models to do significantly better with the correct land cover. The simple representation of BNF in most of these models is based on superseded results. Since the relationship between, say NPP and BNF, is weak, we would not expect any significant improvement with an improvement in vegetation type, except by chance. The main results are from the global models and global upscaled observations, and while interesting, the local comparisons are supporting evidence.

We cover this limitation in the Discussion, starting on line 229.

*The authors should also present the land-only and future scenario simulations in the methodology section (section 2.1).*

Added.

*Moreover, the authors should explain the choice of ssp370 as the future scenario. Why don't you analyze other future scenarios?*

We chose SSP3-7.0 because it had the greatest number of model simulations available, and the other SSPs had fewer model simulations available. Given that the number of ensemble members is already small, a cross-comparison might lead to biased results.

Nevertheless, it appears unlikely that there is a strong non-linearity of the response to climate change for BNF in the models that would lead to strongly alter conclusions about the uncertainty in BNF projections in the current CMIP6 ensemble. Therefore, we judge that one future scenario will provide the most important conclusions without cluttering the paper with unnecessary results that show gradations of the same results with fewer models.

*Specific comments*

*Line 21: should be Eyring et al., 2016 instead of Taylor et al., 2012.*

Done.

*Line 120: SI Table 1 is not available in the supplement link of the manuscript.*

Apologies, this should have been added to the SI during the upload process. It is now included.

*Figure 1: "T The global" remove the initial "T"*

Done.

*Figure 1: add information about the panels in the caption.*

Done.

*Figure 1: the second row has the name of the model truncated due to the figure caption. Reshape the figure.*

Done.

*Lines 139-140: "which […] tropical forest BNF". Rephrase.*

Done.

*SI Figure 2: second line in the caption "CM5" should be "CLM5"*

Done.

*SI Figure 2: Why do you compute the values over the 2005-2014 period, while the reference period is 1980-2014 in the paper?*

Apologies, this is a typo and has been corrected.

*SI Figure 3: explain the grey lines of panel a in the caption.*

Done.

*Figure 3, line 484: "reported by the paper". Which paper?*

Clarified that this means the paper the observational data came from.

*Figure 4: panels c, d and e are missing.*

Apologies, in the conversion from draft to Biogeosciences formatting these panels were missed off. They have been re-added.

*Lines 261-262: "Some models […] nitrogen cycle". Rephrase.*

Done.

---

## Author Comment (AC2)

**Response to reviewer 2**

We thank the reviewer for their time in reviewing this paper. The reviewer comments are in *blue italics* and our responses are in plain black text.

*I can only make few editorial comments on this manuscript:*

*135-136: confusion in the numbering of figures 1b, 1c, 1d.*

Fixed.

*144-145: in figure 3 the units of measurement are in kgN, and in the text - in kg.*

Clarified to kgN.

*152-153: the marker "b" is missing in Figure 3.*

Done.

*187-200: figures 4c, 4d, 4e are missing in the manuscript.*

In the conversion from draft to Biogeosciences formatting this part of the figure was missed off. Our apologies: it is now reattached.

*470-…: we see BNF for the periods 1980-2014 (Figure 1), 1980-2015 (Figure 2), 2005-2014 (Figure SI 2). May be it would be better to average BNF estimates for the period of 1980-2014.*

Apologies, this is a typo and has been corrected.

*I think this is a very good student-level technical study. I do not see new scientific results in this manuscript. For these reasons, I consider this article to be inappropriate for Biogeosciences status.*

We disagree with the notion that this study does not contain new scientific results and would therefore be inappropriate for Biogeosciences. The inclusion of the nitrogen cycle is one of the key innovations of CMIP6 compared to CMIP5. BNF is one of the most important factors determining the long-term trajectory of N availability and therefore the N limitation of the simulated carbon cycle. While individual assessments of individual model components have been published elsewhere, This paper is, to our knowledge, the first evaluation of the performance of these new CMIP6 models using a new synthesis of field observations and upscaled observational data.

Evaluation of climate and Earth System Models is a well-known line of scientific enquiry and has been for decades. However, routine benchmarking tools do not account for any N cycle specific database, and our manuscript thereby addresses a key gap in the literature. We hope that the demonstration of the current state of N cycle modelling in Earth System Models contributes to improved consideration of the ecological complexity of BNF in these models.

---

## Author Response (AR2)

Thank you to the reviewers for their time and consideration. Reviewer comments are in *blue italics*, and our responses are in plain black.

*Reviewer 1*

*Specific comments*

*Lines 66-67: "As an additional check on the performance of the ESMs, we also looked at the BNF of a number of land surface models (LSMs) used in the ESMs presented here, and include this in the SI." Add explicitly that these runs are offline simulations performed with the set of LSMs.*

Added.

*Line 80: no reference for TaiESM1?*

No reference was available at the time of writing, but has now been added.

*Line 95: no reference for AWI-ESM?*

No reference was available at the time of writing, but has now been added.

*Line 141: "estimates is in" the "is" should be removed*

Thank you, corrected.

*Figure 4a,b: I don't see the AWI-ESM circles.*

They aren't there because AWI-ESM did not have that simulation available.

*Lines 195-196: ESMs taking part in the C4MIP effort should have used the same nitrogen deposition forcing coming from CCMI as described in Jones et al., 2016.*

We have added this reference.

*Lines 201-202: I don't see where these numbers are listed.*

They are listed as changes to single number ratios in brackets on the next line.

*Line 226: Discussion should be section 4 instead of 5*

Fixed.

*Line 249: "based Attempts" the "A" should be "a"*

Fixed.

*Figure 1, lines 502-503: inversion of low latitude and high latitude definition. Low latitude should be between 30°S and 30°N, while high latitude more than 60°N or 60°S.*

Thank you for catching that, we have corrected it.

*References:*

*Jones, C. D., Arora, V., Friedlingstein, P., Bopp, L., Brovkin, V., Dunne, J., Graven, H., Hoffman, F., Ilyina, T., John, J. G., Jung, M., Kawamiya, M., Koven, C., Pongratz, J., Raddatz, T., Randerson, J. T., and Zaehle, S.: C4MIP – The Coupled Climate–Carbon Cycle Model Intercomparison Project: experimental protocol for CMIP6, Geosci. Model Dev., 9, 2853–2880, https://doi.org/10.5194/gmd-9-2853-2016, 2016.*

*Reviewer 2*

*It seems to me that the question about the scientific significance raised by Reviewer #2 was not adequately addressed in the revised manuscript. The article evaluates the ability of some CMIP6 ESM to reproduce the observed rates of biological nitrogen fixation and reports the results which are new and deserving publication.*

We thank the reviewer for their time in reviewing and acknowledgement that these findings are new and deserving of publication.

*The question is whether the results should be published in Biogeosciences or elsewhere. "Does the manuscript represent a substantial contribution to scientific progress within the scope of this journal (substantial new concepts, ideas, methods, or data)?" This question should be somehow answered in the article (preferably in an indirect way). The authors reply to Reviewer #2 comment does not look very convincing. I would recommend to highlight the conclusion that evaluated models "does not have explanatory power for variations in net primary productivity or the coupled nitrogen-carbon cycle" and therefore cannot be used for predicting biological nitrogen fixation response to global change.*

We're grateful to reviewer for highlighting the potential for misunderstanding of the quoted conclusion from the abstract, and have amended the sentence appropriately to clarify that it means the variation within model ensemble. And while the reviewer is correct that the lack of relationship between NPP and BNF means that models with BNF based on NPP should not be used for projections of BNF response to global change, this conclusion does not hold for process-based models. This is a nuanced conclusion that indicates the importance of model structural uncertainty, and we have adjusted the text accordingly.

*The title should be changed to convey the major result of the study. The current title forms a wrong impression that paper is not to report any new findings.*

That is a wrong impression that we agree must be corrected. To that end, we have re-titled the paper: Assessment of the Impacts of Biological Nitrogen Fixation Structural Uncertainty in CMIP6 Earth System Models.

*Perhaps, this can be done by minor revision if handling associate editor decides that the article is "appropriate to Biogeosciences status" in principle. (N.B. Do not forget to insert heading "3 Results" above the subheading "3.1 Present day BNF").*

Our apologies if the lack of a Results heading made the paper more difficult to understand; it has now been added.